# SENP2 restrains the generation of pathogenic Th17 cells in mouse models of colitis

Tsan-Tzu Yang[1,2], Ming-Feng Chiang[2], Che-Chang Chang [3], Shii-Yi Yang[2], Shih-Wen Huang[4], Nan-Shih Liao[4], Hsiu-Ming Shih [5], Wei Hsu [6] & Kuo-I Lin [1,2✉]

The molecular mechanisms contributing to the regulation of Th17-mediated inflammation remain underexplored. We here report a SUMO-specific protease (SENP)2-mediated pathway induced in pathogenic Th17 cells that restricts the pathogenesis of inflammatory colitis. SENP2 regulates the maturation of small ubiquitin-like modifiers (SUMO) and recycles SUMO from the substrate proteins. We find higher levels of SENP2 in pathogenic Th17 cells. By deleting *Senp2* in T-cell lineages in mice, we demonstrate that the lack of *Senp2* exacerbates the severity of experimental colitis, which is linked to elevated levels of GM-CSF$^+$IL-17A$^+$ pathogenic Th17 cells and more severe dysbiosis of the intestinal microbiome. Adoptive transfer experiments demonstrate the cell-autonomous effect of *Senp2* in restraining Th17 differentiation and colitis. The enzymatic activity of SENP2 is important for deSUMOylation of Smad4, which reduces Smad4 nuclear entry and *Rorc* expression. Our findings reveal a SENP2-mediated regulatory axis in the pathogenicity of Th17 cells.

[1] Graduate Institute of Immunology, College of Medicine, National Taiwan University, Taipei 10002, Taiwan. [2] Genomics Research Center, Academia Sinica, Taipei 11529, Taiwan. [3] The Ph.D. Program for Translational Medicine, College of Medical Science and Technology, Taipei Medical University, Taipei 11031, Taiwan. [4] Institute of Molecular Biology, Academia Sinica, Taipei 11529, Taiwan. [5] Institute of Biomedical Sciences, Academia Sinica, Taipei 11529, Taiwan. [6] Forsyth Institute, Harvard School of Dental Medicine, Harvard Stem Cell Institute, Harvard University, Cambridge, MA 02142, USA. ✉email: kuoilin@gate.sinica.edu.tw

nflammatory bowel disease (IBD) is a chronic inflammatory disorder and a risk factor for developing colorectal cancer. Many etiological factors are linked to IBD, including genetic factors, impaired homeostasis of mucosal immunity, and disturbed gut microbiota[1,2]. It has been shown that Th1 and Th17 subsets contribute to the pathogenesis of IBD[3] and that there was altered post-translational modification (PTM) by small ubiquitin-like modifier (SUMO) modification in the colonic epithelium of a mouse model of ulcerative colitis[4]. However, the role of SUMOylation in T cells and IBD pathogenesis remains unexplored.

SUMO modifies lysine (K) residues of target proteins. The SUMO family consists of four members: SUMO-1, SUMO-2, SUMO-3, and SUMO-4[5]. SUMOylation regulates many aspects of cellular functions. It involves the maturation of SUMO precursors by SUMO-specific proteases (SENPs), and covalent conjugation of the mature form of SUMOs to the ε-amino group of a K residue within the consensus sequence (ψKXE) through the cascade of enzymatic action of a specific activating enzyme (E1), conjugating enzyme 9 (E2; Ubc9), and ligase (E3)[6,7]. The process of SUMOylation is dynamic and can be reversed by SENPs. There are six genes encoding SENP family proteins, namely, SENP1, SENP2, SENP3, SENP5, SENP6, and SENP7, which deconjugate SUMO proteins[8], but the role of SENP family proteins in the immune system is still poorly understood.

Th17 cells can be stimulated by transforming growth factor-β (TGF-β) and the IL-6 or IL-23p40 pathway[9]. TGF-β signaling interrupts the Smad4-dependent suppression of *Rorc* (encoding RORγt) via a mechanism of triggering the degradation of a transcriptional repressor, SKI, thereby reversing the Smad4/SKI interaction-mediated repression and allowing the induction of Th17 cells[10]. Th17 cells can be classified into non-pathogenic and pathogenic cells, depending on the presence of cytokines in the microenvironment. TGF-β1/IL-6-mediated signaling induces non-pathogenic Th17 cells, whereas TGF-β3/IL-6 or IL-6/IL-23/IL-1β-mediated signaling induces pathogenic Th17 cells[11]. In addition to IL-17A, pathogenic Th17 cells often co-produce IFNγ; as a result, pathogenic Th17 cells may promote the differentiation of Th1 cells in inflamed tissues[12]. Pathogenic Th17 cells also produce granulocyte macrophage-colony stimulating factor (GM-CSF), which is critical for the pathogenicity of Th17 cells in inflammation[13,14]. Given that whether PTM by SUMOylation regulates the generation of pathogenic or non-pathogenic Th17 is still unknown, and that among SENP family members, only SENP2 can deSUMOylate Smad4[15], we here generated T-cell-specific *Senp2*-knockout mice to study the effect of disruption of SUMO cycling on the T-cell lineages. We found that SENP2 plays a negative role in pathogenic Th17 cell generation.

## Results

### Generation of T-cell-specific *Senp2*-deficient mice
We generated T-cell-specific *Senp2*-knockout mice to examine whether disruption of SUMO cycling plays a role in T-cell homeostasis, differentiation, and T-cell-mediated diseases. Mice with Cre-mediated conditional deletion of *Senp2* in T cells were generated by crossing *Senp2*[f/f] mice carrying two loxP sites flanking *Senp2* exon 4[16] with Lck-Cre mice (Supplementary Fig. 1a). Splenic pan-T cells were isolated from the *Senp2*[f/f] × Lck-Cre[+] mice (hereafter called CKO) and control littermates (wild-type, WT). Genomic PCR (Supplementary Fig. 1b), genomic qPCR (Supplementary Fig. 1c), RT-qPCR (Supplementary Fig. 1d), and immunoblotting (Supplementary Fig. 1e) results demonstrated the efficacy of *Senp2* deletion in T cells.

Early T-cell development was disturbed in mice lacking *Senp1*[17]. We first examined whether T-cell development was

affected in CKO mice by analyzing the distribution of thymocyte subsets. However, the frequencies of the single-positive, double-positive (DP), and double-negative (DN) thymocytes, which are divided into DN1 (CD25[−]CD44[+]), DN2 (CD25[+]CD44[+]), DN3 (CD25[+]CD44[−]), and DN4 (CD25[−]CD44[−]), as well as the natural regulatory T cells (nTregs), were comparable in the thymus of WT and CKO mice (Supplementary Fig. 2a–d). The homeostasis of CD4 and CD8 T cells in the spleen, cervical lymph node (cLN), or mesenteric lymph node (MLN) of CKO and WT mice also showed no difference (Supplementary Fig. 2e, f). It was noted that, in a steady state, more IFNγ[+]CD4[+] and IL-17A[+]CD4[+] T cells were found in the spleen and cLN of CKO mice (Fig. 1a–c). Therefore, unlike the defects caused by deletion of *Senp1*[17], lack of *Senp2* in T cells did not disturb T-cell development or CD4 and CD8 T-cell homeostasis. However, SENP2 appeared to prevent the differentiation of IFNγ[+]CD4[+] and IL-17A[+]CD4[+] T cells in a steady state.

We next examined whether SENP2 is differentially expressed in Th subsets. Splenic naive CD4[+] T cells were subjected to the differentiation of various Th subsets, including Th1, Th2, Treg, TGF-β-induced non-pathogenic Th17, and IL-1β-induced pathogenic Th17 cells (Fig. 1d). Notably, Th1 and Th17 cells express significantly higher levels of *Senp2* mRNA and protein than Th0 cells (Fig. 1e, f). Furthermore, pathogenic Th17 cells have highly elevated levels of *Senp2* mRNA and protein compared with non-pathogenic Th17 cells (Fig. 1e, f). These results suggest a potential role of SENP2 in Th17 and/or Th1-mediated pathogenesis.

### *Senp2* deficiency promotes the progression of inflammatory colitis
Chemically induced colitis is a widely used approach for studying the T-cell-mediated pathogenesis of IBDs[18]. Previous studies reported that Th1 cells and Th17 cells contribute to the chronic progression of dextran sodium sulfate (DSS)-induced colitis, which damaged mucosal epithelial cells and caused inflammation in the colon[19]. Therefore, we tested whether *Senp2* deficiency in T cells affects the progression of DSS-induced colitis (Fig. 2a). Remarkably, we found that the survival rate was significantly lower in CKO mice than that in WT mice after DSS treatment (Fig. 2b). Consistent with this, the loss of SENP2 caused more severe colitis as the length of the colon, measured at day 14 after DSS induction, decreased in CKO mice, while WT and CKO mice had similar colon lengths after $H_2O$ treatment (Fig. 2c). The enhanced severity of colitis in CKO mice was further confirmed by histological evaluation of the colon tissues (Fig. 2d, e), characterized by increased infiltration of mucosal immune cells, an elevated level of colonic epithelial hyperplasia, and enhanced thickening of the muscularis mucosae. The overall clinical score was significantly increased in CKO mice after DSS induction (Fig. 2f).

Furthermore, the frequency of Th1 and Th17 cells in the spleen, lamina propria, or MLN of DSS-treated WT or CKO mice was analyzed on day 14 post-DSS induction. More IFNγ[+] and IL-17A[+]CD4[+] T cells were found in MLN of CKO mice (Fig. 2g, h). In particular, the frequency of IL-17A[+]CD4[+] T cells in the MLN of DSS-treated CKO mice was significantly higher than that in DSS-treated WT mice (Fig. 2h). However, the frequency of FOXP3[+] Treg cells was comparable in the MLN of DSS-induced WT and CKO mice (Fig. 2g, h). Similarly, more IL-17A[+]CD4[+] T cells, but a comparable frequency of Treg cells, were found in the lamina propria of DSS-induced CKO mice (Fig. 2i, j). These findings are consistent with increased production of IL-17A in the supernatant harvested from MLN of DSS-induced CKO mice and phorbol 12-myristate 13-acetate (PMA)/ionomycin-stimulated pan-T-cell culture derived from MLN of DSS-induced CKO mice (Fig. 2k). We noted that a significantly increased proportion of

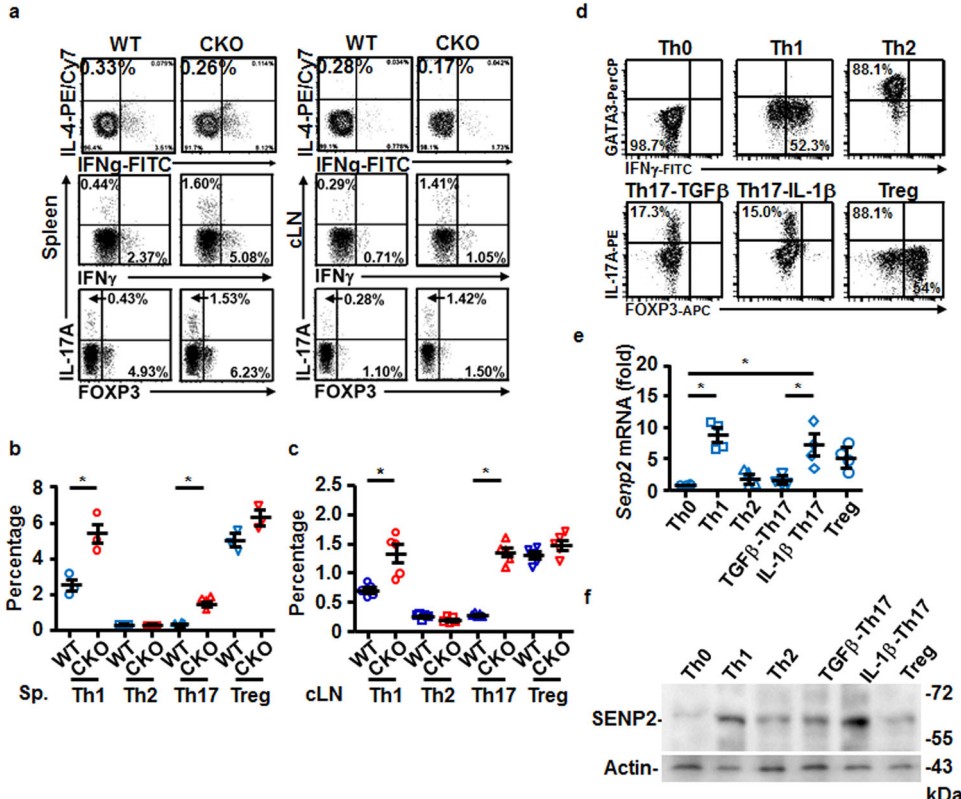

**Fig. 1 Altered frequency of IL-17A+ CD4 T-cell subset in CKO mice in a steady state. a–c** FACS showing the frequency of IFNγ+ or IL-17+ CD4+ T cells in the spleen and cLN of WT and CKO mice (**a**), and the statistical analysis of indicated CD4+ subsets in the spleen (**b**) and cLN (**c**) of WT and CKO mice. **d** FACS showing the frequency of various indicated Th subsets differentiated in vitro on day 5. **e, f** The *Senp2* mRNA (**e**) and protein (**f**) levels on day 5 of each indicated Th subset culture. Results in (**e**) are normalized to *actin* mRNA and then compared with Th0. Results in (**b, c, e**) are mean ± SD ($n = 3-5$). Statistical analysis was done by Student's *t* test in (**b, c**) and one-way ANOVA in (**e**). *$P < 0.05$.

IL-17A+CD4+ cells in the lamina propria and MLN of DSS-induced CKO mice co-expressed IFNγ (Fig. 2l, m). To further test whether the production of pathogenic Th17 cells was altered in DSS-treated CKO mice, we examined the frequency of GM-CSF-producing Th17 cells. There were more pathogenic Th17 cells (GM-CSF+IL-17A+CD4+ T cells) found in MLN of DSS-induced CKO mice, compared with that in DSS-induced WT mice (Fig. 2n, o). The production of pathogenic Th17-associated cytokine, GM-CSF, was higher in the supernatant harvested from MLN of DSS-induced CKO mice and in PMA/ionomycin-stimulated pan-T-cell culture derived from MLN of DSS-induced CKO mice (Fig. 2p). Taken together, these results suggested that the progression of DSS-induced colitis was enhanced in the absence of *Senp2* in T cells, which is linked to the increased generation of pathogenic Th17 cells.

**Th17 cells contribute to more severe colitis in CKO mice.** To further delineate the cell-autonomous effect of SENP2 on Th17 cells, we conducted adoptive transfer experiments. Transfer of naive CD4+ T cells into immunodeficient Rag2-knockout (*Rag2*$^{-/-}$) mice induces colitis and small bowel inflammation[20,21]. Using this method (Fig. 3a), we detected a worse survival rate and an increased clinical score for the adoptive transfer of CKO naive CD4+ T cells into *Rag2*$^{-/-}$ mice (Fig. 3b, c), compared with those of the WT naive CD4+ T cells. Histopathologically, the colon of *Rag2*$^{-/-}$ mice transferred with CKO naive CD4+ T cells showed more severe immune cell infiltration (Fig. 3d) and thicker muscularis mucosae (Fig. 3e) than that from *Rag2*$^{-/-}$ mice transferred with WT naive CD4+ T cells. *Rag2*$^{-/-}$ mice transferred with PBS alone exhibited

normal colon histology (Fig. 3d). Furthermore, the frequency of IL-17A+CD4+ T cells, but not IFNγ+CD4+ T cells and Treg cells, in MLN and lamina propria of *Rag2*$^{-/-}$ mice transferred with CKO naive T cells was higher than that in *Rag2*$^{-/-}$ mice transferred with WT naive T cells (Fig. 3f, g). Similarly, co-transfer of WT naive CD4+ T cells (CD45.1) together with CKO naive CD4+ T cells (CD45.2) into *Rag2*$^{-/-}$ mice (Fig. 3h) also gave rise a higher frequency of IL-17A+CD4+ (CD45.2) T cells than IL-17A+CD4+ (CD45.1) T cells in *Rag2*$^{-/-}$ mice (Fig. 3i, j). As our prediction, co-transfer of equal ratio of CD45.1 and CD45.2 naive WT CD4+ T cells produced comparable frequency of IL-17A+CD4+ T cells in the *Rag2*$^{-/-}$ recipient mice (Fig. 3i, j). These combined data demonstrated the cell-intrinsic effect of SENP2 on inhibiting Th17 cell generation.

To further verify the connection between SENP2 in Th17 cells and colitis, we generated another *Senp2*-knockout mouse model able to delete the gene of interest in thymocytes and Th17 cells by crossing *Rorc*-Cre mice[22] with *Senp2*$^{f/f}$ (hereafter called Rorc-CKO) mice. The Rorc-CKO mice also exhibited an enhanced Th17 cell population in the spleen and cLN in a steady state (Supplementary Fig. 2g), compared with the WT mice. Notably, the Rorc-CKO mice displayed a worse survival rate after induction of DSS using the protocol described in Fig. 2a than WT mice (Fig. 3k). Moreover, the frequency of IL-17A+CD4+ T cells was significantly higher in the MLN of DSS-induced Rorc-CKO mice than that in DSS-induced WT mice (Fig. 3l, m). Our results again show that the presence of *Senp2* in Th17 cells reduces colitis.

**The intestinal bacterial composition is altered in DSS-treated CKO mice.** The imbalance of Th17 and Treg cells was reported to

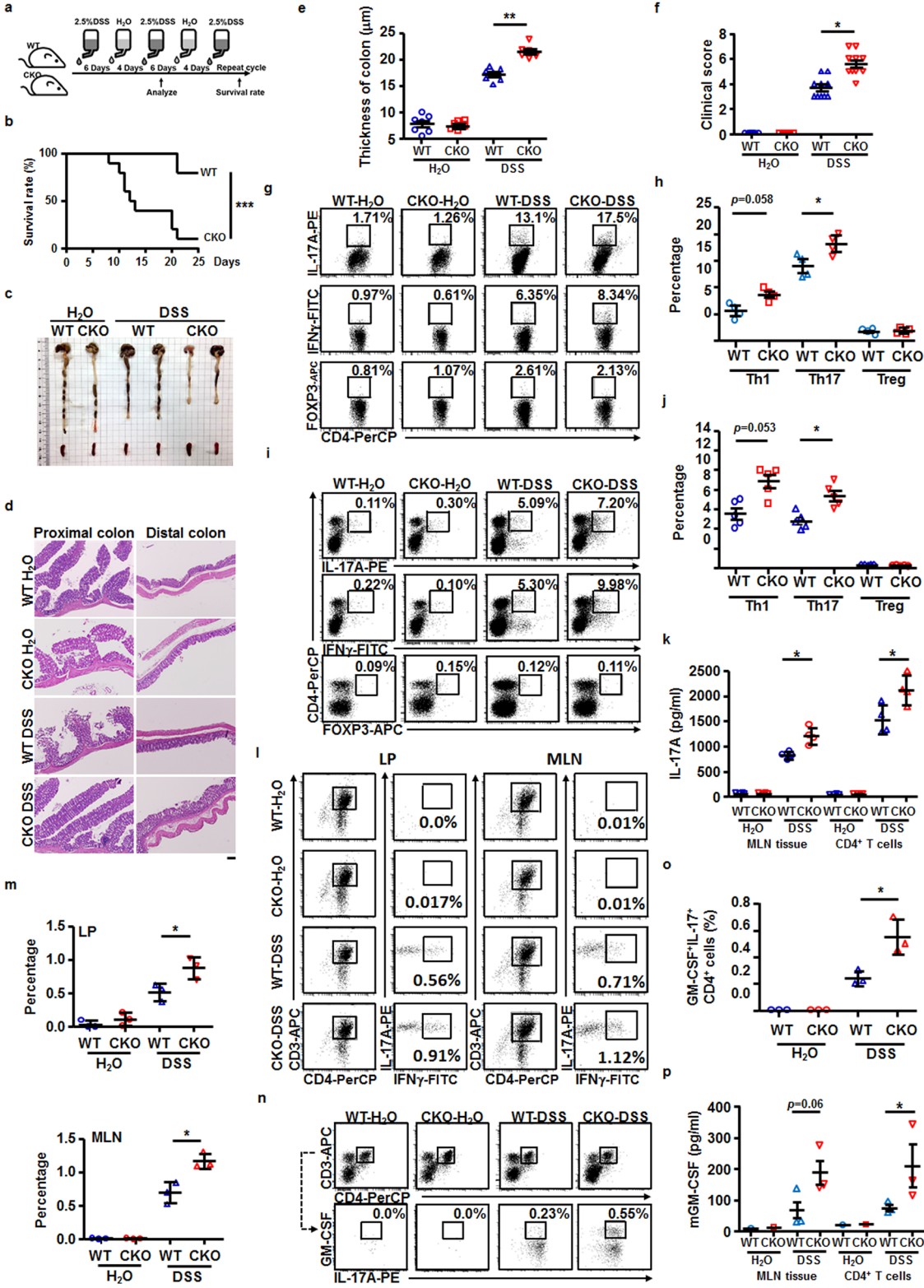

be correlated with the dysbiosis of the gut microbiome[23]. We thus analyzed the composition of the gut microbiota in a steady state and DSS-treated co-housed WT and CKO mice. Results from principal component analysis (PCA) of the composition of intestinal microbiota in the CKO and WT mice in a steady state as well as in the DSS-induced colitis showed that *Senp2* deficiency in T cells did not alter the general gut microbiome in a steady state (Fig. 4a, b), while samples from DSS-induced WT and CKO

mice had distinct clusters, indicating an association of alteration of intestinal microbiota caused by the deletion of *Senp2* and more severe inflammation (Fig. 4c, d). Furthermore, WT and CKO mice after DSS treatment had distinct differences in the abundances of individual genera of gut microbes (Fig. 4c), while the overall bacterial composition in a steady state was similar between WT and CKO mice (Fig. 4b). Specifically, we found that the abundances of *Escherichia-Shigella* and *Clostridium* were

**Fig. 2 Senp2 deficiency promotes the progression of DSS-induced colitis. a** Condition of DSS-induced colitis model in this study. **b** The survival rate of DSS-treated mice was monitored every day and analyzed using Kaplan–Meier statistics. $n = 10$ in each group. **c** Colon length and spleen size were measured in CKO and WT mice at day 14 after $H_2O$ or DSS induction. **d**, **e** The histopathology of the colon in $H_2O$- or DSS-treated WT and CKO mice was examined by H&E staining at day 14. The scale bar represents 50 μm. **f** The quantitative clinical score of WT and CKO mice at day 14 after $H_2O$ or DSS treatment. **g–j** FACS showing the frequency of Treg (bottom in (**g**, **i**)), Th1 (middle in (**g**, **i**)), and Th17 (top in (**g**, **i**)) cells harvested from MLN (**g**) or lamina propria (**i**) of the indicated mice at day 14. Statistical analysis is shown in (**h**, **j**). **k** ELISA showing the production of IL-17 in the tissue supernatant of MLN of indicated mice, and PMA/ionomycin-stimulated CD4[+] T cells isolated from MLN of indicated mice at day 14. **l**, **m** FACS showing the frequency of IFNγ[+]IL-17A[+]CD4[+] T cells in lamina propria (LP, left) or MLN (right) of indicated mice at day 14. The statistical analysis of IFNγ[+]IL-17A[+] CD4[+] cell frequency in LP (top) or MLN (bottom) is shown in (**m**). **n**, **o** FACS showing the frequency of IL-17A[+]GM-CSF[+] cells in the CD4[+] gate in MLN of indicated mice at day 14 (**n**), and the statistical analysis was shown (**o**). **p** The production of GM-CSF from the tissue supernatant of MLN of indicated mice at day 14, and indicated PMA/ionomycin-stimulated CD4[+] T cells was analyzed by ELISA. Results in (**e**, **f**, **h**, **j**, **k**, **m**) are mean ± SD ($n = 4$−$10$). Each dot represents one mouse. Statistical analysis was done by Student's $t$ test. *$P < 0.05$, **$P < 0.005$.

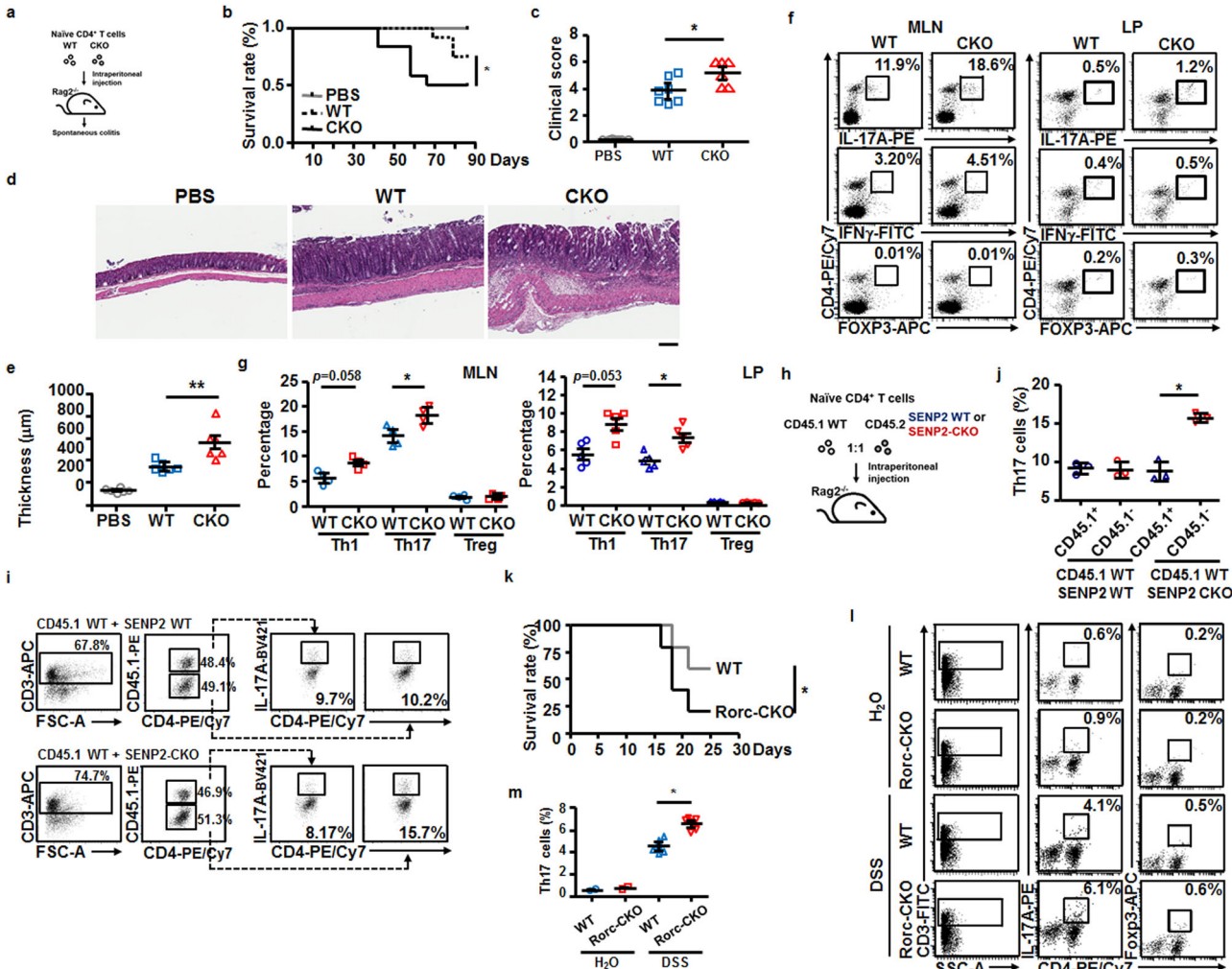

**Fig. 3 Elevated levels of Th17 cells contribute to the exacerbated colitis in the absence of Senp2. a** Illustration showing the procedures of adoptive transfer of $4 \times 10^5$ naive splenic CD4[+] T cells from WT or CKO mice, or PBS to $Rag2^{-/-}$ mice. **b** The survival rates of adoptively transferred $Rag2^{-/-}$ mice given PBS, naive CD4[+] WT, or naive CD4[+] CKO cells. Results were analyzed by Kaplan–Meier statistics ($n = 8$ in each group). **c** The quantitative clinical scores of $Rag2^{-/-}$ recipients at day 49 after adoptive transfer. **d** H&E staining showing the histopathology of the colon in the recipient $Rag2^{-/-}$ mice at day 49 after adoptive transfer. The scale bar represents 50 μm. **e** The thickness of the colon was measured by ImageScope. **f**, **g** FACS showing the frequency of Treg (bottom), Th1 (middle), and Th17 (top) cells in the CD3[+] gate in MLN of $Rag2^{-/-}$ recipients at day 49 after adoptive transfer (**f**). Statistical analysis of the frequency of Th1, Th17, and Treg cells in (**f**) is shown in (**g**). **h** Illustration showing the procedures of adoptive co-transfer of CD45.1 WT CD4[+] T cells and CD45.2 WT or CD45.2 CKO CD4[+] T cells into $Rag2^{-/-}$ mice. **i** FACS showing the frequency of CD45.1[+] Th17 and CD45.1[−] Th17 cell populations in the MLN of indicated recipient mice at day 14, and statistical analysis of Th17 cell frequency is shown in (**j**). **k** The survival rate of WT and Rorc-CKO mice subjected to DSS-induced colitis using the protocols as described in Fig. 2a. Results were analyzed using Kaplan–Meier statistics. $n = 7$ in each group. **l**, **m** FACS showing the frequency of Th17 (middle) and Treg (right) cell populations in MLN of indicated mice at day 14 (**l**), and statistical analysis of Th17 cell frequency is shown in (**m**). Results in (**c**, **e**, **g**, **j**, **l**) are mean ± SD ($n = 3$-$5$). Each dot represents one mouse. Statistical analysis was done by Student's $t$ test. *$P < 0.05$, **$P < 0.005$.

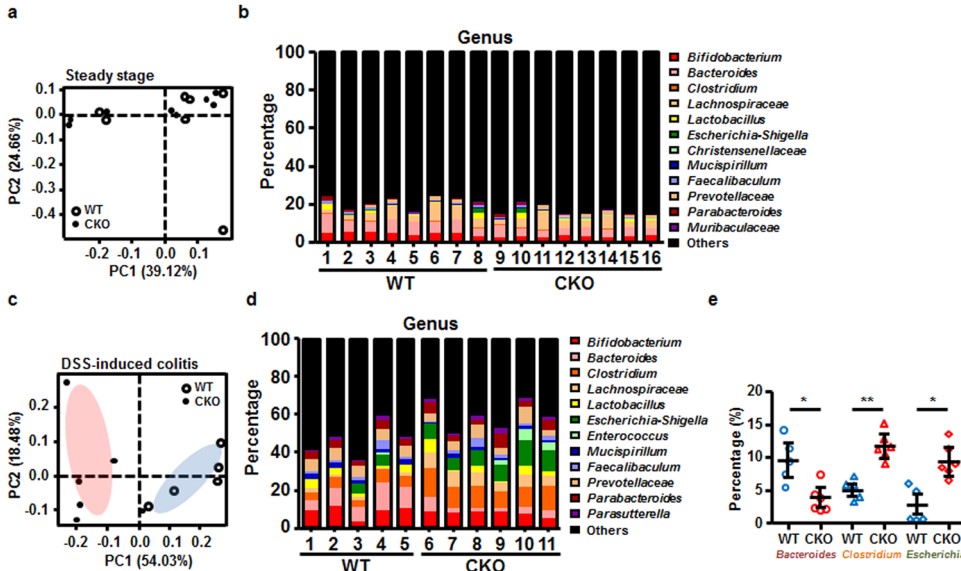

**Fig. 4 The gut bacterial composition was altered in DSS-treated Senp2-deficient mice. a, b** 16 S RNA sequencing of the fecal microbiota of WT and CKO mice in a steady state (**a**) and on day 14 after DSS induction (**b**). Results were analyzed by principal component analysis (PCA). **c, d** The fecal bacterial composition in a steady state (**c**) and in DSS-treated (**d**) WT and CKO mice at the genus level. **e** The levels of *Bacteroides*, *Clostridium*, and *Escherichia-Shigella* in the fecal microbiota from DSS-treated WT and CKO mice on day 14. Results are mean ± SD (*n* = 8 in WT and CKO mice in (**a**, **b**); *n* = 5 in WT and *n* = 6 in CKO in (**c–e**)). Statistical analysis was done by Student's *t* test. *P < 0.05, **P < 0.005).

increased, while the abundance of *Bacteroides* was reduced, in DSS-induced CKO mice (Fig. 4e). Previous studies suggested that *Bacteroides fragilis* modulated host-microbial symbiosis by suppressing the differentiation of Th17 cells[24–26]. Therefore, the reduction of *Bacteroides* in DSS-treated CKO mice supports our findings of elevated generation of Th17 cells in DSS-treated CKO mice.

**The catalytic activity of SENP2 is required for the inhibition of pathogenic Th17 differentiation.** To elucidate the mechanism underlying SENP2-mediated differentiation of Th17 cells, we examined the ex vivo generation of pathogenic Th17 cells affected by the loss of SENP2. There were more pathogenic Th17 cells (IL-1β-induced) differentiated from CKO naive CD4+ T cells than from WT naive CD4+ T cells, while the levels of non-pathogenic Th17 cells (TGF-β induced) differentiated from WT and CKO naive CD4+ T cells were comparable (Fig. 5a, b). We also detected higher levels of Th1 cells differentiated from CKO naive CD4+ T cells (Fig. 5a, b). In addition, we observed higher levels of pathogenic Th17 cells differentiated from splenic naive CD4+ cells of Rorc-CKO mice (Supplementary Fig. 2h). Other Th cell subsets appear to differentiate normally from Rorc-CKO naive CD4+ cells (Supplementary Fig. 2h). Therefore, we assumed that SENP2 mainly restrains the generation of pathogenic Th17 cells. The proliferation and cell death of CKO and WT naive CD4+ T cells in response to anti-CD3 and anti-CD28 stimuli, as well as to the condition favoring the pathogenic Th17 polarization, were comparable (Supplementary Fig. 3a–d), suggesting that the SENP2-mediated pathway may not be involved in the expansion of pathogenic Th17 cells.

To rule out the possibility that the abovementioned phenotypes related to the enhanced production of pathogenic Th17 cells from CKO mice might result from minor differences in T-cell development or activation, we crossed *Senp2*f/f mice with mice carrying the inducible estrogen receptor/cre (ERcre) gene in all tissues[27]. The resulting inducible *Senp2*-knockout (CKO-ER) mice displayed inducible deletion of *Senp2* in pathogenic Th17-polarizing CD4+ T cells after 4-hydroxytamoxifen (4-OHT)

treatment as genomic PCR results showed that *Senp2* levels started to decline in the 4-OHT-treated CKO-ER T cells from day 2 in pathogenic Th17 polarizing culture (Fig. 5c). Notably, the frequencies of IFNγ+ and IL-17A+ cells were elevated in the 4-OHT-treated CKO-ER pathogenic Th17 culture compared with those from the 4-OHT-treated WT culture (Fig. 5d, e). As pathogenic Th17 cells co-express IFNγ and IL-17A[28], our results implied an intrinsic effect of *Senp2* on the negative regulation of pathogenic Th17 differentiation.

To further demonstrate the requirement of the enzymatic activity of SENP2 for pathogenic Th17 differentiation, we transduced pathogenic Th17 polarizing culture derived from WT or CKO mice with lentiviral vectors expressing GFP-tagged wild-type SENP2, GFP-tagged catalytic domain mutant SENP2-C/S, or control (Ctrl) vector (Fig. 5f). The catalytic triad of residues in SENP2 includes Cys548, His478, and Asp495;[29] thus, Cys548 replaced by Ser548 represents the enzymatically dead SENP2 mutant form, SENP2-C/S[15]. We found that transducing SENP2-expressing vector into the CKO cells effectively inhibited the production of IL-17A+ cells, and was associated with levels of Th17 differentiation similar to those of the culture derived from WT mice transduced with Ctrl vector. Note that CKO cells expressing GFP-SENP2-C/S retained increased levels of Th17 differentiation, in comparison with CKO cells expressing SENP2 (Fig. 5g, h). Furthermore, the frequency of IL-17A+ T cells was comparable between CKO cells with Ctrl vector transduction and with SENP2-C/S vector transduction (Fig. 5g, h). These results indicate that the enzymatic activity of SENP2 is essential for the negative regulation of pathogenic Th17 differentiation.

**SENP2 modulates Smad4 SUMOylation in pathogenic Th17 cells.** Previous studies suggested that STAT3 can be SUMOylated at the K541 residue after IL-6 stimulation. The de-conjugation of SUMO by SENP3 was also shown to promote the transcriptional activity of STAT3[30]. Given that the STAT3 signaling pathway is required for early Th17-cell differentiation and *Rorc* transcription[31,32], and that STAT3 sustains Th17-cell proliferation and cytokine production[33], we examined the status of STAT3

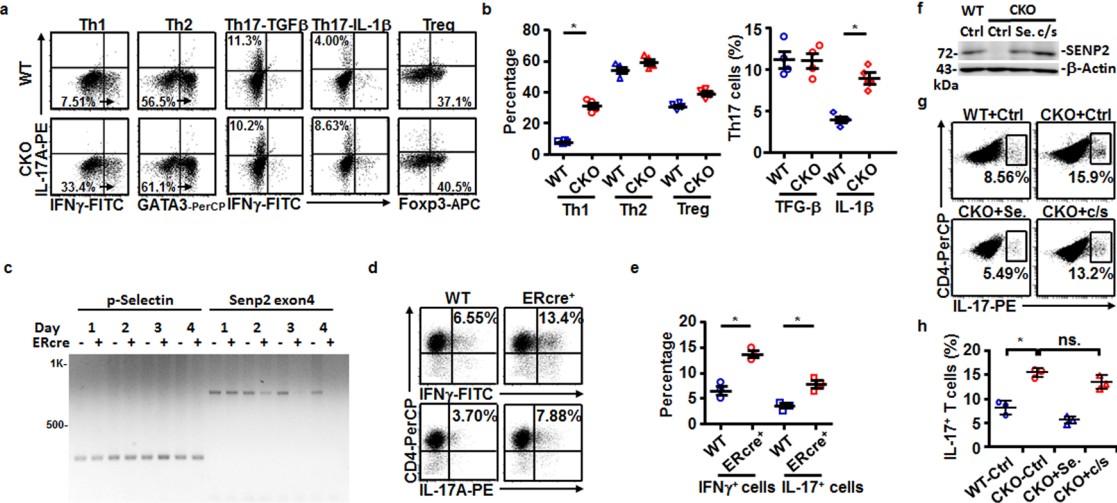

**Fig. 5 Increased frequency of generation of pathogenic Th17 cells in culture in the absence of Senp2. a, b** FACS showed the production of pathogenic Th1 and Th17 cells by differentiating naive CD4+ T cells from WT and CKO mice for 5 days (**a**). Statistical analysis is shown in (**b**). **c** Genomic PCR showing deletion of the *Senp2* allele in the presence of 500 nM 4-OHT in differentiating pathogenic Th17 cultures derived from CKO-ER (*Senp2*f/f ERcre+) mice at the indicated days, compared with those from WT-ER (*Senp2*f/f ERcre−) mice. *P-Selectin* was used as an internal control. **d, e** FACS showing the frequency of IFNγ+ (top) and IL-17A+ (bottom) cells at day 5 of pathogenic Th17 polarizing culture derived from CKO-ER+ and WT mice with the addition of 4-OHT (**d**). Statistical analysis is shown in (**e**). **f–h** Re-introduction of wild-type SENP2 (Se) or catalytic domain mutant of SNEP2 (c/s) via GFP carrying lentiviral vector into differentiating pathogenic Th17 culture derived from either WT or CKO mice. Immunoblotting showing the expression of exogenous SENP2 in CKO culture transduced with vector alone (Ctrl), wild-type SENP2 (Se.), or c/s mutant of SENP2 at 48 h after transduction (**f**). FACS showing the frequency of IL-17A+ cells after re-introduction of Ctrl, wild-type SENP2 (Se.), or c/s SENP2 in differentiating pathogenic Th17 culture derived from either WT or CKO mice (**g**). Statistical analysis is shown in (**h**). Results in (**b, e, h**) are mean ± SD (*n* = 3). Statistical analysis was done by Student's *t* test. *P < 0.05. ns. not significant.

activation in WT and CKO pathogenic Th17 polarizing cultures. However, we found that the activation of STAT3, characterized by phosphorylated STAT3 at the Y705 site, was similar in WT and CKO cultures soon after pathogenic Th17 polarization (Supplementary Fig. 4a). We also examined a co-transcriptional factor, Smad4, that has been reported to bind directly to *Rorc* and regulate the non-pathogenic Th17 differentiation[10], although its role in pathogenic Th17 differentiation remained unclear. The SENP2 protein segment containing residues 363−589 has also been shown to participate in the Smad4 interaction and deSU-MOylation in 293T co-transfectants[15]. Using purified recombinant GST-SENP2$^{363−589}$ segment (Supplementary Fig. 4b) and lysates prepared from pathogenic Th17 cells, we showed that GST-SENP2$^{363−589}$ interacts with Smad4 in a GST pull-down assay (Fig. 6a). We then examined the levels of SUMO-1-conjugated Smad4 in WT and CKO polarizing pathogenic Th17 cultures by denaturing immunoprecipitation (IP), followed by immunoblotting analyses. The results showed that a higher level of SUMO-1-conjugated Smad4 was detected in CKO pathogenic Th17 cells compared with that in the WT culture (Fig. 6b). However, the total Smad4 levels in WT and CKO polarizing pathogenic Th17 cultures remained similar. This indicates that the lack of *Senp2* in pathogenic Th17 cultures resulted in the accumulation of SUMO-1-conjugated Smad4.

Smad4 can be regulated by SUMOylation at site K159[34,35]. We were able to confirm Smad4 SUMOylation at K159 by co-IP using lysates prepared from 293T cells co-transfected with Flag-tagged wild-type Smad4 or Flag-tagged SUMO conjugation-defective Smad4 (Smad4 K159R) and GFP-tagged SUMO-1. Indeed, Smad4 K159R failed to be covalently modified by GFP-SUMO-1 (Fig. 6c). We next tested whether SUMO-1 modification affects the nuclear translocation of Smad4. The vector expressing GFP-SUMO-1 along with the vector expressing Flag-Smad4 K159R, or Flag-Smad4 wild-type were co-transfected into IL-6-stimulated EL4 CD4 T cells (Supplementary Fig. 4c), followed by monitoring

Smad4 nuclear translocation using confocal microscopy and DAPI staining as a nuclear marker. Our results showed that SUMO-defective Smad4 is retained in the cytosol better than wild-type Smad4 (Fig. 6d, e). Consistently, higher levels of Smad4 protein (Fig. 6f) were detected in the nuclear extracts isolated from CKO pathogenic Th17 culture as compared with those from WT pathogenic Th17 culture, suggesting that SUMOylated Smad4 translocates into the nucleus more efficiently, leading to enhanced transactivation activity. Consistent with this, we observed higher levels of *Rorc* mRNA in IL-6-stimulated EL4 cells expressing Smad4 wild-type than in those expressing Smad4 K159R (Fig. 6g). This result was linked with the increased enrichment of Smad4 binding to the known binding site on *Rorc* promoter region[10], as shown by the chromatin immunoprecipitation (ChIP) using chromatin prepared from WT and CKO pathogenic Th17 cultures and anti-Smad4 antibody (Fig. 6h). Accordingly, higher levels of RORγt protein were found in the polarizing pathogenic Th17 culture derived from splenic CKO mice (Supplementary Fig. 5a, b), and in IL-17A+CD4+ T cells of DSS-induced mice with *Senp2* deletion (Fig. 6i). Our combined results showed that SENP2-mediated deSUMOylation of Smad4 restricts Smad4 nuclear localization and reduces *Rorc* transcription.

## Discussion

Dysregulation of Th17 responses underlies multiple inflammatory diseases[36]. In this study, we demonstrate that *Senp2* deficiency promotes the pathogenesis of inflammatory colitis using two knockout mouse models in which *Senp2* is deleted in developing thymocytes or mature Th17 cells. The pathogenic Th17 cells are also affected by the absence of *Senp2* in inflammatory colitis. We infer that SENP2 is critical for pathogenic Th17 regulation as the expression of *Senp2* in the pathogenic Th17 polarizing culture is elevated compared with that in naive CD4+ T cells and non-pathogenic Th17 polarizing culture. It has been suggested that *SENP2* is the direct target of NF-κB induced by DNA damage[37].

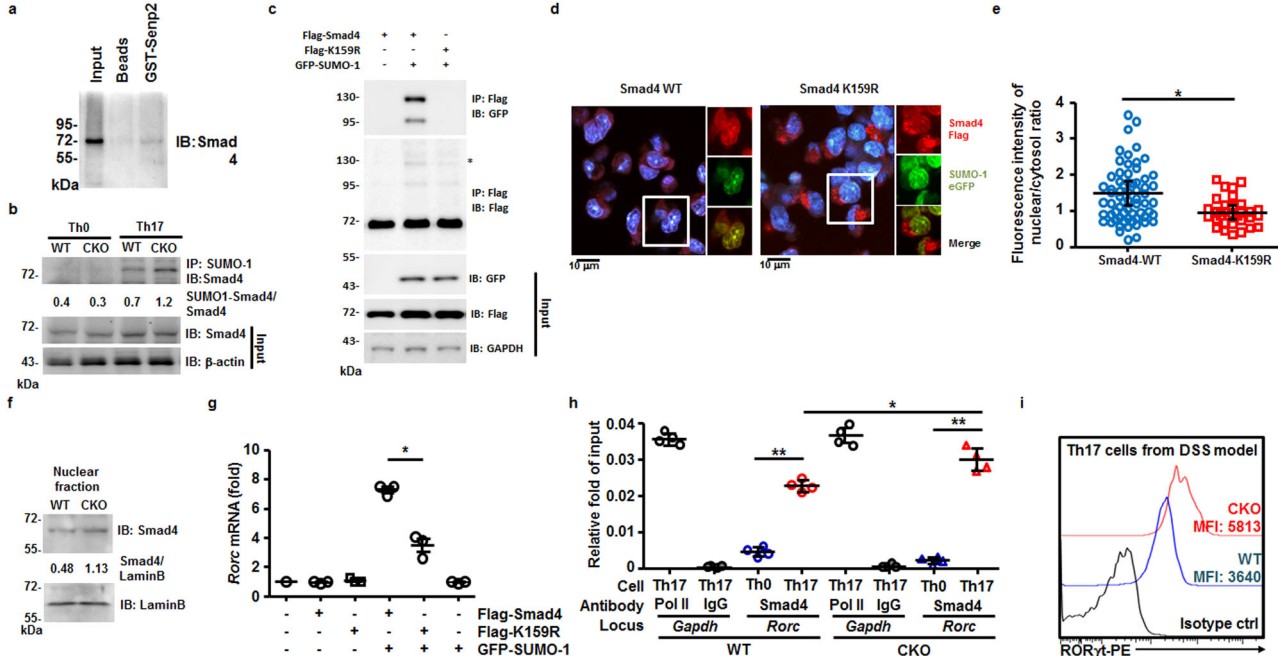

**Fig. 6 SENP2 modulates the level of SUMO-1-conjugated Smad4 in pathogenic Th17 cells and Smad4 nuclear localization. a** Immunoblotting (IB) showing that Smad4 protein in day 5 lysates isolated from polarizing pathogenic Th17 cells was pulled down by recombinant GST-SENP2[363–589] fusion protein segment and Glutathione Sepharose 4B beads, but not by beads alone. **b** Denaturing immunoprecipitation (IP) using the anti-SUMO-1 antibody in IP showing the levels of SUMO-1-conjugated Smad4 in Th0 and day 5 pathogenic Th17 cell culture derived from WT and CKO mice. The ratios of protein band intensity of SUMO-1 conjugated Smad4 in IP vs. Smad4 in input were indicated. **c** Co-IP examining the conjugation of GFP-SUMO-1 with wild-type Smad4, or with Smad4 K159R mutant, in 293 T cells transfected with GFP-tagged SUMO-1 and Flag-tagged Smad4 or Flag-tagged Smad4 K159R. GAPDH was used for internal control blotting. **d**, **e** Immunofluorescent staining showing the nuclear levels of WT Smad4 or Smad4 K159R in IL-6 (50 ng/ml)-treated EL4 T cells co-transfected with GFP-SUMO-1 at day 2 after transfection (**d**). Results were examined by confocal microscopy after staining with an anti-Flag antibody (red). DAPI (blue) was used as a nuclear marker. Statistical analysis of the ratio of red signals co-expressed with blue signals in transfected EL4 cells from two independent experiments is shown in (**e**). **f** IB showing the levels of Smad4 in the nuclear extracts prepared from the day 5 culture of pathogenic Th17 cells derived from WT and CKO mice. The ratios of protein band intensity of Smad4 vs. Lamin B were indicated. **g** RT-qPCR of *Rorc* mRNA levels in IL-6-treated EL4 cells transfected with vectors expressing GFP-tagged SUMO-1 and Flag-tagged Smad4 or Flag-tagged Smad4 K159R. β-actin mRNA was used for normalization. **h** ChIP assay showing the levels of Smad4 enriched on *Rorc* locus in day 5 pathogenic Th17 cells polarized from naive WT or CKO CD4⁺ T cells. qPCR was used to quantify the fold changes of Pol II binding (positive control) or Smad4 binding on the indicated gene loci of indicated cells between immunoprecipitated and input samples. IgG was used as the negative control in the IP step. **i** The mean fluorescence intensity (MFI) of RORγt levels from FACS analysis of the CD4⁺IL-17A⁺ cells in MLN of DSS-treated WT and CKO mice at day 14. Results are mean ± SD ($n = 3$ in (**g**) and 4 in (**h**)). Statistical analysis was done by Student's t test. *$P < 0.05$.

An NF-κB-binding sequence has been shown to be located 70 bp upstream of the transcription start site of *Senp2*[38]. Given that both canonical and noncanonical NF-κB pathways are involved in the generation of inflammatory T cells[39], NF-κB likely contributes to the enhanced transcription of *Senp2* in pathogenic Th17 cells. Although Th1 cells are not significantly increased in CKO mice after DSS induction and lack of SENP2 in Rorc-CKO naive CD4 T cells did not affect the differentiation of Th1 cells in culture, this cannot rule out the role of cell-intrinsic SENP2 in Th1-cell differentiation. Whether SENP2 in Th1 cells is also involved in regulating the progression of Th1-dominant diseases in vivo remains elusive.

The frequency of pathogenic Th17 cells, but not non-pathogenic ones, is increased in polarizing CKO culture. To our knowledge, this is the first evidence supporting the mechanism of selectively restraining pathogenic GM-CSF-producing Th17 differentiation. How does SENP2 negatively regulate pathogenic Th17 differentiation? The enzymatic activity of SENP2 is required for restraining pathogenic Th17 generation as the re-introduction of the enzymatically deficient form of SENP2 (SENP2-C/S) fails to reduce the generation of pathogenic Th17 cells in CKO culture. In addition, Smad4 is an important functional substrate of SENP2 in pathogenic Th17 cells. The deSUMOylation of Smad4 reduces

the nuclear localization of Smad4 in T cells. Thus, pathogenic Th17 cells lacking *Senp2* have higher levels of SUMO-1-conjugated Smad4, thereby permitting more effective entry of Smad4 into the nucleus. It has been reported that Smad4 interacts with a transcription repressor, SKI, to suppress *Rorc* transcription. The stimulation of TGF-β signaling leads to SKI degradation, thereby activating the Th17 gene expression signature[39]. Consistent with a recent report[40], the SKI levels are also reduced under the condition of pathogenic Th17 polarization, despite the absence of TGF-β stimulation (Supplementary Fig. 5c). However, the levels of SKI in pathogenic Th17 cell cultures derived from naive CD4 WT and CKO T cells are comparable (Supplementary Fig. 5c), indicating that SUMOylation of Smad4 did not affect the degradation of SKI under pathogenic Th17 polarization. Furthermore, febrile temperature-induced pathogenicity of Th17 cells depends on heat shock response and signaling mediated by the SUMOylation pathway, specifically through the regulation of Smad4 SUMOylation[41]. Our findings support this notion that more nuclear translocation of SUMOylated Smad4 is linked to the enhanced pathogenicity of Th17 cells.

In the model of DSS-induced colitis, the abundance of *Bacteroidetes* and *Bacteroidales* is decreased, while that of *Gammaproteobacteria*, *Enterobacteriales*, and *Escherichia-Shigella*, among

**Fig. 7 The proposed mode of action of SENP2 in preventing excessive inflammation in pathogenic Th17 cells.** Induction of SENP2 by inflammatory cytokines in pathogenic Th17 cells serves as a regulatory modulator to tune down the SUMOylation and nuclear translocation of Smad4, thereby preventing the excessive production of IL-17A and GM-CSF.

others, is increased, which might result in the re-establishment of gut microbiota equilibrium[42]. Our results support the changes in those intestinal bacteria after DSS induction. More dramatic changes of *Escherichia-Shigella*, *Clostridium*, and *Bacteroides* in the gut of CKO mice after DSS induction may reflect the more severe colitis phenotypes. The gut microbiome is comparable in WT and CKO mice in a steady state, despite the frequency of Th17 cells being increased in the spleen and cLN of CKO mice in a steady state. This suggests that overall CD4$^+$ T cell responses may not be significantly altered under steady state, and that elevated Th17 frequency in CKO mice in a steady state may not be pathologic. We also speculate that the IL-17A$^+$CD4$^+$ T cells found in CKO mice in a steady state may be the precursors of pathogenic Th17 cells, which will become GM-CSF$^+$IL-17A$^+$CD4$^+$ pathogenic Th17 cells upon inflammatory insults.

In summary, *Senp2* deficiency in T cells exacerbates IBD, which is linked to enhanced pathogenic Th17-cell differentiation. SENP2 deficiency does not affect STAT3 activation and SKI degradation, but SENP2-mediated deSUMOylation of Smad4 restricts Smad4 nuclear localization and *Rorc* transcription in pathogenic Th17 cells (Fig. 7). Therefore, SENP2 in pathogenic Th17 cells serves as a negative regulator to tone down the Smad4-mediated transcription, revealing it as a selective target to modulate pathogenic Th17 cells and related inflammatory diseases.

## Methods

**Mice.** *Senp2*$^{f/f}$ mice were described as previously[16]. T cell-specific *Senp2* knockout mice (CKO, *Senp2*$^{f/f}$Lck$^{Cre+/+}$) and control mice (WT, Lck$^{Cre+/+}$) in C57BL/6 genetic background were raised and kept in the Academia Sinica SPF animal facility, according to the rule of the Animal Protection Act of Taiwan. Lck-Cre mice, *Rag2*$^{-/-}$ mice and CD45.1 mice were from Jackson Laboratory. The experimental protocol was approved by Academia Sinica's Institutional Animal Care and Utilization Committee. In some experiments, *Senp2*$^{f/f}$ mice were bred with R26-ERCre mice (Jackson Laboratory) or RORγ-Cre mice (Jackson Laboratory) in C57BL/6 background. The deletion of *Senp2* was examined by genomic qPCR using 5′-TCCAGCTTCTCCAAGAAACCTAACC-3′ and 5′-CTCATGACC ATTAGTGTGCAGTGCT-3′ primers. Actin was used as the internal control and the sequences are 5′-CATTGCTGACAGGATGCAGAAGG-3′ and 5′-TGCTGGA AGGTGGACAGTGAGG-3′. In some experiments, C57BL/6 mice were purchased from National Laboratory Animal Center, Taiwan. All mice used for experiments were at 8–12 weeks of age.

**Dextran sulfate sodium (DSS)-induced colitis model.** To establish the murine model of DSS-induced colitis, mice were treated with 2.5% DSS (molecular mass

36,000–50,000 kDa, Tseng Hsiang Life Science Ltd.) in water for 6 days, followed by treatment with fresh water for 4 days, after which this cycle was repeated before sacrifice. Mouse colon samples were fixed in 10% (vol/vol) phosphate-buffered formalin, embedded in paraffin, and cut into 3-µm-thick sections for H&E staining. The colon length was measured by ImageScope (Leica Biosystems). In some experiments, 4 × 10$^5$ naive CD4$^+$ T cells, isolated using MACS® (Miltenyi Biotec #130-104-453) from WT and CKO mice, were adoptively transferred into *Rag2*-knockout mice (Jackson Laboratory). The survival, weight loss, stool consistency, and rectal bleeding of mice with colitis were monitored daily and the clinical score of colitis was evaluated in accordance with a previous report:[43] weight loss: 1–5% = 1, 5–10% = 2, 10–15% = 3, >15% = 4; stool consistency: normal stool = 0, loose stool = 1, watery diarrhea = 2; and rectal bleeding: negative = 0, visible blood on feces = 1, an abundance blood around the anus = 2.

**Adoptive transfer and co-transfer experiments.** T-cell adoptive transfer was performed according to a previous study[44]. Briefly, splenic naive CD4$^+$ T cells were isolated from 8-week-old WT and CKO mice. 4 × 10$^5$ naive CD4$^+$ T cells were resuspended in 100 µl of PBS and adoptively transferred by i.p. injection into 8-week-old *Rag2*$^{-/-}$ recipient mice. In co-transfer experiments, 2 × 10$^5$ WT naive CD4$^+$ T cells (CD45.1) were mixed with 2 × 10$^5$ WT naive CD4$^+$ T cells (CD45.2) or with 2 × 10$^5$ CKO naive CD4$^+$ T cells (CD45.2), and adoptively transferred by i.p. injection into 8-week-old *Rag2*$^{-/-}$ recipient mice. The procedures of examining mouse survival, clinical score and histopathology were conducted as those described in DSS-induced colitis model. Two weeks after adoptive transfer, mice were sacrificed and stained with antibodies for FACS analysis.

**Enzyme-linked immunosorbent assay (ELISA).** IL-17A and mGM-CSF ELISA Kit (R&D Systems, Inc.) were used in this study. The experiments were conducted in accordance with the manufacturer's instructions. Briefly, 96-well microplates were coated with 2 µg/ml capture primary antibody in coating buffer containing PBS and incubated overnight at 4 °C. The plates were blocked with 5% BSA in PBS for 2 h at room temperature (RT). The plates were then washed three times with PBS-T (1% Tween 20). Mouse tissue lysates or cultured supernatants were added into coated wells and incubated at RT for 2 h. After three washes with PBS-T, horseradish peroxidase (HRP)-conjugated secondary antibody was added and incubated at RT for another 2 h. The substrate was added and the absorbance was measured using a microplate reader (SpectraMax® M2) at 450 nm.

**Gut microbiota analysis.** Gut microbiota was analyzed in accordance with a previous report[45]. DNA from the colonic feces of DSS-untreated or -treated mice on day 14 was extracted using a WelPrep DNA Kit (Welgene Biotech Co., Ltd., Taipei, Taiwan; Cat no. D001) or QIAamp® DNA Stool Mini Kit (Qiagen, Valencia, CA, USA), in accordance with the manufacturer's instructions, to obtain an OD$_{260/280}$ ratio between 1.8 and 2.0. The V3–V4 hypervariable region of 16 S rDNA was amplified using the bacterial-specific forward (5′-TCGTCGGCAGCGTCAGATGTGTATAAGAGAC AGCCTACGG GGGCGCAG-3′) and reverse (5′-GTCTCGTGGGCTCGGAGATG TGTATAAGAGACAGGACTACGGGTATCTAATCC-3′) primer set with the 16 S Metagenomic Sequencing Library Preparation Kit (Illumina, San Diego, CA, USA). Indexed adapters were added to the amplicons using the Nextera XT Index Kit

(Illumina, San Diego, CA, USA), in accordance with the manufacturer's instructions. The accuracy of amplified DNA size was checked using the 4200 TapeStation System (Agilent Technologies, Santa Clara, CA, USA). After library construction, samples were mixed with MiSeq Reagent Kit v3 (600-cycle) and loaded onto a MiSeq cartridge; then, a $2 \times 300$ bp paired-end sequencing run was performed using the MiSeq platform (Illumina). Sequencing and data analysis were performed by Welgene Biotech Co., Ltd. (Taipei, Taiwan), using Illumina Sequencing-by-Synthesis (SBS) technology.

**Cell lines and culture of primary T cells**. EL4 cells were maintained in RPMI 1640 supplemented with 10% FBS (BenchMark), penicillin (100 U/ml), and streptomycin (100 µg/ml). 293T cells were maintained in DMEM supplemented with 10% FBS (BenchMark), penicillin (100 U/ml), and streptomycin (100 µg/ml). Splenic naive CD4$^+$ T cells were isolated with the isolation kit from MACS® (Miltenyi Biotec. #130-104-453). The purity of naive CD4 T cells (CD3$^+$CD4$^+$CD44$^-$CD62L$^+$) used for culture in this study is >88%. The naive CD4 T cells were cultured in RPMI 1640 supplemented with 10% FBS (Bench-Mark), 2-ME (50 µM), penicillin (100 U/ml), and streptomycin (100 µg/ml), and treated with the combination of following recombinant proteins or antibodies for polarizing various Th subsets: Th1: anti-mouse CD3 antibody (5 µg/ml, eBioscience), anti-mouse CD28 antibody (2 µg/ml, eBioscience), IL-2 (5 ng/ml, PeproTech), IL-12 (10 ng/ml, PeproTech) and anti-mouse IL-4 (30 ng/ml, Pepro-Tech); Th2: anti-mouse CD3 antibody (5 µg/ml, eBioscience), anti-mouse CD28 antibody (2 µg/ml, eBioscience), IL-2 (5 ng/ml, PeproTech), IL-4 (30 ng/ml, PeproTech), anti-mouse IL-12 (10 ng/ml, PeproTech) and anti-mouse IFN-γ antibodies (5 µg/ml, BioLegend); Treg: anti-mouse CD3 antibody (5 µg/ml, eBioscience), anti-mouse CD28 antibody (2 µg/ml, eBioscience), IL-2 (5 ng/ml, PeproTech), hTGF-β (1 ng/ml, PeproTech), anti-mouse IL-12 (10 ng/ml, Pepro-Tech), anti-mouse IL-4 (30 ng/ml, PeproTech) and anti-mouse IFN-γ antibodies (5 µg/ml, BioLegend); non-pathogenic Th17: anti-mouse CD3 antibody (5 µg/ml, eBioscience), anti-mouse CD28 antibody (2 µg/ml, eBioscience), IL-6 (50 ng/ml, PeproTech), hTGF-β (1 ng/ml, PeproTech), anti-mouse IL-12 (10 ng/ml, Pepro-Tech), anti-mouse IL-4 (30 ng/ml, PeproTech) and anti-mouse IFN-γ antibodies (5 µg/ml, BioLegend); pathogenic Th17: anti-mouse CD3 antibody (5 µg/ml, eBioscience), anti-mouse CD28 antibody (2 µg/ml, eBioscience), IL-6 (50 ng/ml, PeproTech), IL-1β (10 ng/ml, PeproTech), IL-23 (10 ng/ml, R&D), anti-mouse IL-12 (10 ng/ml, PeproTech), anti-mouse IL-4 (30 ng/ml, PeproTech) and anti-mouse IFN-γ antibodies (5 µg/ml, BioLegend). In some experiments, splenic naive CD4$^+$ T cells isolated from Senp2$^{f/f} \times$ ERCre$^+$ or Senp2$^{f/f} \times$ ERCre$^-$ mice were treated with 500 nM 4-OHT (Sigma-Aldrich) at day 0 under the condition of pathogenic Th17 polarization.

**Flow cytometric analysis**. Splenic CD4$^+$ T cells were stimulated with PMA (50 ng/ml) and ionomycin (1 mg/ml) (Sigma-Aldrich) in the presence of monensin (eBioscience) for 5 h and then stained with the following antibodies: APC/Cy7-conjugated viability dye (Life Technologies), PE/Cy7-conjugated anti-mouse CD4 (clone number: RM45, BioLegend), PerCP5.5-conjugated anti-mouse CD8 (clone number: 53-5.8, BioLegend) and PE-conjugated anti-mouse CD45.1 (clone number: A20, BioLegend). After fixation and permeabilization by eBioscience™ Foxp3/Transcription Factor Staining Buffer Set (Invitrogen™), cells were stained with FITC-conjugated anti-mouse IFNγ (clone number: XMG1.2, BioLegend), PE/Cy7-conjugated anti-mouse IL-14 (clone number: 11B11, eBioscience), PerCP5.5-conjugated anti-mouse GATA3 (clone number: 16E10A23, BioLegend), PE-conjugated anti-mouse RORγt (clone number: B2D, eBioscience), PE-conjugated anti-mouse IL-17A (clone number: TC11-18H10, BioLegend), BV421-conjugated anti-mouse IL-17A (clone number: N49-653, BioLegend), BV421-conjugated anti-mouse GM-CSF (clone number: MP1-22E9, BD Biosciences), APC-conjugated anti-mouse CD3 (clone number: 145-2c11, BD Biosciences), FITC-conjugated anti-mouse CD3 (clone number: 145-2c11, BD Biosciences), PerCP5.5-conjugated anti-mouse CD4 (clone number: RM45, BD Biosciences), PerCP5.5-conjugated anti-mouse GATA3 (clone number: 16E10A23, BioLegend),PE/Cy7-conjugated anti-mouse IL-4 (clone number: 11B11, eBioscience), PE-conjugated anti-mouse CD62L (clone number: MEL-14, BD Biosciences), APC-conjugated anti-mouse CD44 (clone number: IM7, BD Biosciences), PE/Cy7-conjugated anti-mouse CD25 (clone number:Pc61, BD Biosciences), and APC-conjugated anti-mouse Foxp3 antibodies (clone number: 150D, BioLegend) for 30 min at 4 °C, and washed with PBS three times, followed by FACS analysis with flow cytometry FACScanto (BD Biosciences, San Jose, CA, USA). Results were analyzed using De Novo Software. Gating strategies for flow cytometric analysis in the main figures and supplementary figures are shown in Supplementary Figs. S6–S9.

**RNA isolation and quantitative real-time PCR (RT-qPCR)**. Total RNA was isolated by using the RNeasy mini kit (Qiagen), and 250 ng of RNA was used for cDNA synthesis by using the High-Capacity cDNA Reverse Transcription kit (ABI). qPCR analysis was performed by using Applied Biosystems StepOne™ Real-Time PCR System. The primer sequences for SYBR green real-time PCR are Senp2: 5′-ATGCTGCCAGTTTATTTGGATTC-3′ and 5′-CTGCTGCAGGATCC AACTC-3′, Rorc: 5′-CGCGGAGCAGACACACTTA-3′ and 5′-CCCTGGACCTCTGTTTTGGC-3′, and β-actin: 5′-GCTGTATTCCCCTCCATCGTG-3′ and 5′-CACGGTTG CCTAGG TCAG-3′.

**Immunoblotting**. Harvested cells were lysed with buffer containing 1% Triton X-100, 50 mM Tris (pH 7.5), 10 mM EDTA, 0.02% NaN$_3$, and protease inhibitor cocktail (Sigma). Following one freeze-thaw cycle, the cell lysates were centrifuged at 12,000×g and 4 °C for 20 min. Protein samples prepared by boiling lysates (20 µg) in a sample buffer containing 0.5% (wt/vol) Triton X-100, 20 mM HEPES (pH 7.9), 300 mM NaCl, 1 mM EDTA, and protease inhibitor mixture (Roche) for 5 min were subjected to 8–10% SDS-PAGE analysis. After electrophoresis, the SDS-PAGE gel was then transferred to a PVDF membrane using a semi-dry electro-blotting system. After blocking with 5% skim milk in PBS, the membranes were incubated with primary antibodies at 1:1000 dilution at 4 °C overnight. Primary antibodies used in this study are anti-RORγt (H-190, Santa Cruz), anti-SUMO-1 (D-11, Santa Cruz), anti-Smad4 (D3R4N, Cell Signaling), anti-STAT3-tyr705 (Cell Signaling), anti-Actin-HRP (Gene Script), anti-Flag (Sigma), anti-GFP (Abcam), anti-SENP2 (Abcam or Abgent), anti-Lamin B (B-10, Santa Cruz), and anti-GAPDH (Abcam). The membranes were subsequently washed with 0.05% PBS-Tween 20 and incubated with HRP-conjugated anti-rabbit IgG (Sigma) or anti-mouse IgG (Thermo) secondary antibodies at 1:1000 dilution at RT for 1 h. After washing, the membranes were immersed in ECL (Advansta) solution. The chemiluminescent signal images were captured by ChemStudio touch series, UVP iBox® Explorer Imaging Microscope (Ultra-Violet Products Ltd., Cambridge, UK). Anti-actin and anti-GAPDH blots served as the internal protein loading control. The uncropped blot images in main figures and supplementary figures are provided in Supplementary Figs. S10 and S11, respectively.

**Nuclear extract preparation and Co-immunoprecipitation (IP)**. A total of $1 \times 10^7$ 293T cells transfected with vectors (1.5 µg) expressing Flag-tagged Smad4 or Flag-tagged SUMO site-defective Smad4 (K159R), and green fluorescent protein (GFP)-tagged SUMO-1 at 48 h after transfection were subjected to co-IP analysis, following the procedures described previously[46]. Briefly, cells were lysed by lysis buffer containing 20 mM HEPES (pH 7.9), 20% glycerol, 200 mM NaCl, 1.5 mM MgCl$_2$, 0.5 mM EGTA, 1 mM EDTA, protease inhibitors (Roche, cOmplete™), and 0.5 mM PMSF (Sigma). In some experiments, nuclear extracts were prepared by using NE-PER™ Nuclear and Cytoplasmic Extraction Reagents (Thermo). The lysates were clarified by centrifugation first at 12,000×g at 4 °C for 20 min, followed by IP with anti-Flag antibody (10 µg, Sigma-Aldrich) at 4 °C for 2 h. To denature IP, the lysates (500 µg) prepared from Th0 or pathogenic Th17 cells were denatured by adjusting the solution to final concentrations of 0.5% SDS and 5 mM DTT. The volume of denatured lysates was diluted five times with IP buffer containing 0.2 mM DTT and 0.4% NP-40, and incubated with anti-SUMO-1 antibody-agarose (2 µg per 100–500 µg of protein lysates, Santa Cruz Biotechnology) for IP at 4 °C overnight. Immunoprecipitates from co-IP or denaturing IP were then eluted with $1 \times$ sample buffer and subjected to the immunoblotting analyses.

**GST pull-down**. The preparation of GST-SENP2$^{363-589}$ fusion protein was performed as described previously[15]. A total of $1 \times 10^7$ Th17 cells under pathogenic Th17 polarizing condition for 5 days were lysed with buffer containing 25 mM HEPES (pH 7.5), 150 mM NaCl, 1 mM EDTA, 1 mM EGTA, 0.5% Triton X-100, 1% NP-40 alternative, and protease inhibitor cocktail (Sigma). Approximately 290 µg of lysates were incubated with 10 µg of GST-SENP2$^{363-589}$ fusion protein at 4 °C overnight, and then the lysates were incubated with 30 µL of Glutathione Sepharose 4B beads (Cytiva) at 4 °C for 3 h. After rinsing the beads three times with lysis buffer, the proteins bound to the beads were separated using 10% SDS-PAGE and visualized using Coomassie Blue staining. A total of 2.5 µg of protein lysate harvested from polarizing pathogenic Th17 culture was used as input.

**Lentiviral transduction**. The control lentiviral vector (pLAS3W, Ctrl) and vectors encoding SENP2 (pLAS3W-Senp2) or enzymatically dead SENP2 (pLAS3W-Senp2 c/s) were described in a previous report[15]. Generation of pseudotyped lentivirus was performed essentially as described previously[47]. Targeted primary naive CD4 T cells under pathogenic Th17 polarization were transduced with the lentiviral vectors at day 0 in culture in the presence of 8 µg/ml polybrene (Sigma-Aldrich).

**Confocal microscopy analysis**. EL4 CD4 T cells were maintained in RPMI 1640 supplemented with 10% FBS (BenchMark), penicillin (100 U/ml), and streptomycin (100 µg/ml). Vectors (2 µg) expressing Flag-tagged Smad4 or Flag-tagged K159R Smad4 and GFP-tagged SUMO-1 were co-transfected into EL4 cells ($1 \times 10^6$) by Lipofectamine™ 2000 transfection reagent (Invitrogen™). Cells were harvested at 48 h after transfection and fixed in freshly prepared 4% paraformaldehyde for 30 min at RT, and then transferred into a permeabilization solution containing 0.5% Tween 20 for 30 min at RT. The samples were washed three times with PBS and then incubated for 30 min at 4 °C with the anti-Flag-PE antibody (clone number: L5, Biolegend) and DAPI (Thermo). The sections were washed with PBS on a rotating platform at 4 °C, then mounted in Fluormount G (Southern Biotechnology Associates). The images were acquired by a Leica SP8 confocal microscope. Counting was performed by MetaMorph® Microscopy Automation and Image Analysis Software.

**Chromatin immunoprecipitation (ChIP)**. The ChIP assay was conducted according to the protocol provided by Pierce Magnetic ChIP Kit (Thermo Fisher

Scientific). Pathogenic Th17 cells were polarized from mouse naive CD4$^+$ T cells as described above, and $4 \times 10^6$ Th17 cells were cross-linked with 1% formaldehyde and sonicated to shear genomic DNA. The sample was subjected to immunoprecipitation (IP) overnight at 4 °C with the anti-Smad4 antibody (10 μg) (Cell Signaling), anti-RNA Polymerase II antibody (Thermo Fisher Scientific) as the positive control for its binding to *Gapdh* promoter, and normal rabbit IgG (Thermo Fisher Scientific) as the negative control in the IP step. The protein-chromatin complexes were then purified by magnetic beads (Thermo Fisher Scientific). DNA in the IP was quantified by qPCR analysis. The primer sequences used in ChIP qPCR were: *Rorc*: 5′-GGGGAGAGCTTTGTGCAGAT-3′ and 5′-AGTAGGGTAGCCCAGGACAG-3′, and *Gapdh*: 5′-CATCACTGCCACCCAGAAGACTG-3′ and 5′-ATGCCAGTGAGCTTCCCGTTCAG-3′.

**Cell proliferation and cell death analysis**. Primary naive CD4$^+$ T cells were isolated from the spleen of WT and CKO mice and stained with CFSE (Life Technologies, CellTrace$^{TM}$ cell proliferation kits) at 37 °C for 10 min. CFSE-labeled T cells were cultured with anti-CD3 (5 μg/ml) + anti-CD28 (2 μg/ml) or under conditions favoring pathogenic Th17 polarization. For determining cell death, stimulated cells were labeled with Annexin V and 7-AAD (BD Pharmingen™), in accordance with a previous report[48]. Cell proliferation, as well as cell death, was analyzed using flow cytometry FACScanto (BD Biosciences, San Jose, CA).

**Statistics and reproducibility**. Statistical analysis of the differences in the survival curves of mice with colitis was performed using Kaplan–Meier analysis. Statistical significance was determined by two-tailed Student's *t* tests. $P < 0.05$ was considered statistically significant. One-way ANOVA was used for multiple comparison. Data shown here are the mean ± SD from at least three independent biological experiments, unless otherwise indicated. Statistical analyses were performed using Prism 8 software (GraphPad).

**Reporting summary**. Further information on research design is available in the Nature Portfolio Reporting Summary linked to this article.

## Data availability

All data generated or analyzed during this study are included in this published article and its supplementary information files. 16 S rDNA raw sequencing data reported in Fig. 4 were deposited in NCBI (BioProject accession number: PRJNA972952). Source data for figures can be found in Supplementary Data 1.

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

## Acknowledgements

We thank Dr. Jr-Wen Shui for critically reading this manuscript. We thank Edanz Group (https://en-author-services.edanzgroup.com/ac) for editing a draft of this manuscript. This work was supported by grants from Academia Sinica (AS-IA-107-L05 and AS-VTA-111-04) and National Health Research Institute (NHRIEX108-10835SI).

## Author contributions

T.-T.Y., M.-F.C., C.-C.C., S.-Y.Y. and S.-W.H. performed experiments and analyzed the data. S.-M.S. and W.H. contributed to extensive discussions and interpretation of results. N.-S.L., S.-M.S. and W.H. provided critical materials. T.-T.Y., and K.-I.L. wrote the manuscript. K.-I.L. conceived and supervised research.

## Competing interests

The authors declare no competing interests.
