## [Peer Review File · Communications Biology]

Reviewers' comments:

Reviewer #1 (Remarks to the Author):

In this manuscript, Yang et al showed that Senp2 is required for the suppression of pathogenic Th17 responses. In Senp2 KO mice, the number of IL-17-producing Th17 cells were increased MLN. Senp2 KO mice were highly sensitive to DSS-induced colitis and T cell-dependent colitis models with increased number of IFN-g/IL-17 double producing T cells. The authors then analyzed the mechanism by which Senp2 inhibits pathogenic Th17 responses. Senp2 was shown to associate with and sumoylate Smad4 and inhibits nuclear translocation of Smad4.

Overall, the finding is interesting. However, there are several drawbacks in the study.

1. In Title, IBD should be changed to colitis model, or intestinal inflammation. The authors just used mouse models of IBD.
2. The authors should describe the reason why Senp2 was analyzed in this study among several Senp proteins. In particular, the role of Senp1 in lymphocytes were reported.
3. In Fig. 4, the authors showed dysbiosis in DSS-treated Senp2 CKO mice. However, Senp2 CKO mice developed severe colitis, and it is highly possible that the inflammation caused dysbiosis. Thus, Fig. 4 c-e should be carefully shown or deleted.
4. The authors provided evidence showing the mechanisms by which Senp2 restricts pathogenic Th17 responses. However, all those data are derived from over-expression experiments. The authors should at least show one in vivo or physical evidence, for example, increased nuclear translocation of Smad4 in CD4 T cells of Senp2 CKO mice (immunoblots of cytosolic and nuclear fractions, or immunohist), increased Smad4 binding to the Rorc locus (Chip).

Reviewer #2 (Remarks to the Author):

This is an interesting manuscript, in which the authors show that Senp2 plays an important role for pathogenic Th17 cells. Overall the results are interesting and provide novel insight into the pathways import for the pathogenicity of Th17 cells.

Major points:

- Rorc-Cre was used to delete Senp2 in Th17 cells. However, to my knowledge it is published that this Cre is not specific for Th17 cells, but is active in all T cells (besides ILCs); see Eberl and Littmann 2004. Thus, IL-17A Cre may be more appropriate.
- The authors see difference in Th1 as well as in Th17 cells. Thus, it would be important to discriminate between cell intrinsic and extrinsic effect. To this end the authors could employ a co-transfer experiment in which the inject congenic WT and KO T cells (wt + wt as control) together into Rag KO mice.
- Fig. 2. CD4+ IL-17A+ IFN γ + is an important subpopulation promoting intestinal tissue destruction. What is the frequency of CD4+ IL-17A+ IFN γ + co-producers?
- Fig. 2l. The authors claimed that "there were more pathogenic Th17 cells (GM-CSF+IL-17A+ CD4+ T cells) found in MLN of DSS-induced CKO mice..." However, the GM-CSF+IL-17A+ co-producers are not shown in the figure. Please, add this graph.
- Fig. 3. In the T-cell transfer colitis model (and others), the percentage of Th17 is shown only for MLN. What 's the frequency of Th17 cells in the colon? In all the experimental colitis models used, the

analysis of the colonic T cells is important to understand the influence of these cells in the affected tissue.

- The authors mentioned that "the presence of Senp2 in Th17 cells prevents colitis". However, the WT mice expressing Senp2 also develop colitis (although to a lesser extent) in all the experimental models used. Thus I would soften the conclusion

Minor points:

- Statistical test is missing in some figure legends. Please, indicate if a parametric or non-parametric test is used.

- In the figure legends, the description of each subpanel is sometimes before the indicated subpanel, and sometimes after the indicated subpanel. I would recommend using always the same structure to facilitate the understanding of the figures.

- Please, indicate the number of mice used for each experiment also in the figure legends.

- I would recommend the authors to cite the original article where the findings were described, and not a review article.

-Fig. 6b. A semi-quantitative graph would help a better interpretation of the data.

Reviewers' comments:

Reviewer #1 (Remarks to the Author):

In this manuscript, Yang et al showed that Senp2 is required for the suppression of pathogenic Th17 responses. In Senp2 KO mice, the number of IL-17-producing Th17 cells were increased MLN. Senp2 KO mice were highly sensitive to DSS-induced colitis and T cell-dependent colitis models with increased number of IFN-g/IL-17 double producing T cells. The authors then analyzed the mechanism by which Senp2 inhibits pathogenic Th17 responses. Senp2 was shown to associate with and sumoylate Smad4 and inhibits nuclear translocation of Smad4.

Overall, the finding is interesting. However, there are several drawbacks in the study.
We thank the reviewers for the positive comments and the constructive suggestions.

1. In Title, IBD should be changed to colitis model, or intestinal inflammation. The authors just used mouse models of IBD.

Reply: We thank the excellent suggestion from the reviewer. The title has been modified accordingly to “SEN2 restrains the generation of pathogenic Th17 cells in mouse models of colitis”.

2. The authors should describe the reason why Senp2 was analyzed in this study among several Senp proteins. In particular, the role of Senp1 in lymphocytes were reported.

Reply: Among SENP family members, only SENP2 can deSUMOylate Smad4. Therefore, we investigated into the function of Senp2 in T cells in this manuscript. We thank the reviewer for this thoughtful suggestion, and have included this point in the Introduction in this revision (page 3, lines 34 and 35). Although *Senp1* has been shown to be critical for the development of T and B cells, as shown by the results attached to the right, we did not observe the significant differences in Senp1 mRNA levels (top) and protein levels (bottom) among various differentiating Th subsets in culture. We thus focus on studying Senp2 here.

3. In Fig. 4, the authors showed dysbiosis in DSS-treated Senp2 CKO mice. However, Senp2 CKO mice developed severe colitis, and it is highly possible that the inflammation caused dysbiosis. Thus, Fig. 4 c-e should be carefully shown or deleted.

Reply: We thank the reviewer for this excellent point. One of the reasons showing the microbiota data in Fig. 4 is to demonstrate that *Senp2* deficiency in T cells did not alter the general gut microbiome in a steady state (Fig. 4a and b), suggesting that under the steady state, gut CD4⁺ T cell responses may not be significantly changed. This statement was added in this revised manuscript (page 11, lines 9 and 10). We agree with the reviewer that we should carefully interpret our data related to the changes of microbiome after DSS-induction. We agree that more severe dysbiosis in CKO mice may result from the more severe inflammation as more

Escherichia-Shigella, *Clostridium*, and *Bacteroides* in CKO mice after DSS induction may reflect the more severe colitis phenotypes (page 7, line 3 and page 11, lines 5-7).

4. The authors provided evidence showing the mechanisms by which *Senp2* restricts pathogenic Th17 responses. However, all those data are derived from over-expression experiments. The authors should at least show one in vivo or physical evidence, for example, increased nuclear translocation of Smad4 in CD4 T cells of *Senp2* CKO mice (immunoblots of cytosolic and nuclear fractions, or immunohist), increased Smad4 binding to the *Rorc* locus (Chip).

Reply: We actually showed the higher levels of ROR γ t protein in IL-17A⁺CD4⁺ T cells of DSS-induced mice with *Senp2* deletion in our original submission (Fig. 6g, changed to Fig. 6i in this submission).

In this revision, we conducted two new experiments to address the reviewer's comments:

a. Western blotting of nuclear extracts isolated from pathogenic WT and CKO Th17 culture: we show that more Smad4 in the nuclear extract from CKO Th17 culture (Fig. 6f in this revised manuscript) (page 9, lines 19-21).

b. ChIP using chromatin harvested from pathogenic WT and CKO Th17 culture and anti-Smad4 antibody: we show that increased levels of Smad4 binding to *Rorc* promoter (Fig. 6h in this revised manuscript) (page 9, lines 24-27).

Reviewer #2 (Remarks to the Author):

This is an interesting manuscript, in which the authors show that Senp2 plays an important role for pathogenic Th17 cells. Overall the results are interesting and provide novel insight into the pathways import for the pathogenicity of Th17 cells.

We thank the reviewer for the positive comments.

Major points:

- Rorc-Cre was used to delete Senp2 in Th17 cells. However, to my knowledge it is published that this Cre is not specific for Th17 cells, but is active in all T cells (besides ILCs); see Eberl and Littmann 2004. Thus, IL-17A Cre may be more appropriate.

Reply: Indeed, ROR γ t is expressed in lymphoid tissues and is essential for the development of thymocytes, Th17 cells and other lymphoid tissues (PMID: 10875923, 16990136, etc). We thank the reviewer for raising this important point. We reasoned that SENP2 may not play significant roles in other Th subsets. This is supported by our new experiments by conducting ex vivo Th subset polarization culture using naïve splenic CD4 T cells derived from Th17-CKO and WT mice. Again, we found that higher levels of pathogenic Th17 cells were generated by naïve CD4 T cells isolated from Th17-CKO mice. The new results are provided in the Supplementary Fig. 2h in this revision. However, we observed normal differentiation of other various Th subsets using naïve CD4 T cells derived from Th17-CKO cultures, as compared with WT cultures. We agree with the reviewer's point that Rorc-Cre is not specific to Th17 cells, and decide to change Th17-CKO to Rorc-CKO mice in the entire manuscript of this revision.

-The authors see difference in Th1 as well as in Th17 cells. Thus, it would be important to discriminate between cell intrinsic and extrinsic effect. To this end the authors could employ a co-transfer experiment in which the inject congenic WT and KO T cells (wt + wt as control) together into Rag KO mice.

Reply: To address this question, we co-transferred WT (CD45.1) naïve CD4 T cells together with WT (CD45.2) or with CKO (CD45.2) naïve CD4 T cells (1: 1 ratio) into Rag2 KO mice, and then examined the generation of Th17 cells in the recipients. Again, we observed higher frequency of CD4⁺IL-17A⁺ (CD45.2) T cells in the MLN of Rag2 KO recipients 14 days after adoptive transfer. This result again supports the notion of cell intrinsic role of Senp2 in restraining Th17 generation. Our new results are shown in the Fig. 3 h-j in this revision (page 6, lines 13-20). We thank the reviewer for this excellent suggestion.

- Fig. 2. CD4⁺ IL-17A⁺ IFN γ ⁺ is an important subpopulation promoting intestinal tissue destruction. What is the frequency of CD4⁺ IL-17A⁺ IFN γ ⁺ co-producers?

Reply: In response to the reviewer's suggestion, we conducted new analyses of CD4⁺IL-17A⁺ IFN γ ⁺ co-producers in the lamina propria (LP) and MLN of DSS-treated WT and CKO mice. Our new results are shown in the Fig. 2 l and m in this revision (page 5, lines 25 and 26).

- Fig. 2l. The authors claimed that “there were more pathogenic Th17 cells (GM-CSF+IL-17A+ CD4⁺ T cells) found in MLN of DSS-induced CKO mice...” However, the GM-CSF+IL-17A+ co-producers are not shown in the figure. Please, add this graph.

Reply: We apologize for the ambiguous statement and labels in the Fig. 2l in our original submission, which showed the increased frequency of GM-CSF⁺IL-17A⁺ co-producers in MLN of DSS-induced CKO mice. In this revision, we re-plotted our dot-plots and the new results are shown (Fig. 2n, o in this revision). We thank the reviewer for noting our unclear labels in the figure.

- Fig. 3. In the T-cell transfer colitis model (and others), the percentage of Th17 is shown only for MLN. What's the frequency of Th17 cells in the colon? In all the experimental colitis models used, the analysis of the colonic T cells is important to understand the influence of these cells in the affected tissue.

Reply: We agree. In fact, the frequency of Th17 cells in the lamina propria (LP) of DSS-treated

WT and CKO mice was shown in our previous submission (Fig. 2i, j). In this revision, we added new data further showing the frequency of Th17 cells in the colon lamina propria after adoptive transfer, and in the lamina propria of DSS-treated WT and CKO mice. Specially, we show that the frequency of IL-17A⁺CD4⁺ T cells are increased in colon lamina propria of *Rag2* deficient recipients after transferred with naïve CKO CD4⁺ T cells (Fig. 3f, g in this revision), and the frequency of IFN- γ ⁺IL-17A⁺ co-expressing CD4⁺ T cells are increased in colon lamina propria of CKO mice after DSS induction as compared with that in WT mice after DSS induction (Fig. 2l, m in this revision).

3f

3g

2l

2 m

- The authors mentioned that “the presence of *Senp2* in Th17 cells prevents colitis”. However, the WT mice expressing *Senp2* also develop colitis (although to a lesser extent) in all the experimental models used. Thus I would soften the conclusion

Reply: This is an excellent point. We toned down our statement in the Discussion of this revision on page 11, line 20.

Minor points:

- Statistical test is missing in some figure legends. Please, indicate if a parametric or non-

parametric test is used.

Reply: We thank the reviewer for carefully reading our manuscript. The statistical methods have been added in the figure legends of this revision.

- In the figure legends, the description of each subpanel is sometimes before the indicated subpanel, and sometimes after the indicated subpanel. I would recommend using always the same structure to facilitate the understanding of the figures.

Reply: Following the reviewer's suggestion, we have amended the figure legends to make the statement structure consistent.

- Please, indicate the number of mice used for each experiment also in the figure legends.

Reply: We have indicated the number of mice in each figure legends. We thank the reviewer for pointing out this omission.

- I would recommend the authors to cite the original article where the findings were described, and not a review article.

Reply: We have made the adequate changes (the changed references were indicated in the References, Page 18-20).

-Fig. 6b. A semi-quantitative graph would help a better interpretation of the data.

Reply: The semi-quantitative graph of Fig. 6b is included in this revision. Likewise, semi-quantitative graph of our new results in Fig. 6f is also provided. We thank the reviewer for this thoughtful suggestion.

REVIEWERS' COMMENTS:

Reviewer #1 (Remarks to the Author):

The authors well responded to the reviewer's comments, and the manuscript is now greatly improved.

Reviewer #2 (Remarks to the Author):

The authors have successfully addressed all my comments and concerns.